# Duplication and subfunctionalisation of the general transcription factor IIIA (*gtf3a*) gene in teleost genomes, with ovarian specific transcription of *gtf3ab*

**Iratxe Rojo-Bartolomé[1], Jorge Estefano Santana de Souza[2], Oihane Diaz de Cerio[1], Ibon Cancio[1] ***

**1** CBET Research Group, Centre for Experimental Marine Biology and Biotechnology (PiE-UPV/EHU) and Dept. of Zoology and Cell Biology (Fac. Science and Technology), University of the Basque Country (UPV/EHU), Bilbao, Basque Country, Spain, **2** Bioinformatics Multidisciplinary Environment – BioME, Universidade Federal do Rio Grande do Norte, Natal, Rio Grande do Norte, Brazil

* ibon.cancio@ehu.eus

**Data Availability Statement:** All relevant data are within the manuscript and its Supporting Information files.

## Abstract

Fish oogenesis is characterised by a massive growth of oocytes each reproductive season. This growth requires the stockpiling of certain molecules, such as ribosomal RNAs to assist the rapid ribosomal assembly and protein synthesis required to allow developmental processes in the newly formed embryo. Massive 5S rRNA expression in oocytes, facilitated by transcription factor 3A (Gtf3a), serves as marker of intersex condition in fish exposed to xenoestrogens. Our present work on Gtf3a gene evolution has been analysed *in silico* in teleost genomes and functionally in the case of the zebrafish *Danio rerio*. Synteny-analysis of fish genomes has allowed the identification of two *gtf3a* paralog genes, probably emerged from the teleost specific genome duplication event. Functional analyses demonstrated that *gtf3ab* has evolved as a gene specially transcribed in oocytes as observed in *Danio rerio*, and also in *Oreochromis niloticus*. Instead, *gtf3aa* was observed to be ubiquitously expressed. In addition, in zebrafish embryos *gtf3aa* transcription began with the activation of the zygotic genome (~8 hpf), while *gtf3ab* transcription began only at the onset of oogenesis. Under exposure to 100 ng/L 17β-estradiol, fully feminised 61 dpf zebrafish showed transcription of ovarian *gtf3ab*, while masculinised (100 ng/L 17α-methyltestosterone treated) zebrafish only transcribed *gtf3aa*. Sex related transcription of g*tf3ab* coincided with that of *cyp19a1a* being opposite to that of *amh* and *dmrt1*. Such sex dimorphic pattern of *gtf3ab* transcription was not observed earlier in larvae that had not yet shown any signs of gonad formation after 26 days of oestradiol exposure. Thus, *gtf3ab* transcription is a consequence of oocyte differentiation and not a direct result of estrogen exposure, and could constitute a useful marker of gonad feminisation and intersex condition.

**Funding:** This work has been funded through research projects of MINECO (AGL2012-33477 and AGL2015-63936_R), Basque-Government (PhD fellowship to IRB, S-PE13UN101 & IT810-13). The funders had no role in study design, data collection and analysis, decision to publish, or preparation of the manuscript The funders had no role in study design, data collection and analysis, decision to publish, or preparation of the manuscript.

**Competing interests:** The authors have declared that no competing interests exist.

## Introduction

A great variety of anthropogenic chemicals present estrogen- or androgen-like properties displaying biological activities similar to those of endogenous hormones. Upon bioaccumulation such chemicals interfere with the normal hormonal function altering, for example, the normal control of sexual differentiation, gametogenesis and reproduction. They are known as reproductive endocrine disrupting compounds (reproductive EDCs) [1, 2, 3] and they have become a global concern due to their ubiquity in aquatic environments [2, 4, 5]. EDCs reach the aquatic environment from wastewater treatment plants, and strongly influence fish development, especially in sensitive early life stages even at concentrations as low as parts per trillion [1, 3, 6].

Sex determination systems in teleost fish are extremely plastic but poorly understood with most of the karyotyped species not showing differentiated sex chromosomes [7]. In addition, teleost genomes show a partially duplicated genome after the teleost specific genome duplication event and this further complicates the identification of potential sex linked genes [8]. Both sex determination and differentiation are also environmentally driven in fish. Among others, environmental factors, such as hypoxia, food availability, and temperature [7, 9, 10] have been shown to strongly influence sex determination and differentiation. Sex differentiation and reproduction in fish can also be influenced by exposure to EDCs [2]. Exposure to such compounds during critical periods of development in zebrafish (*Danio rerio*) has been reported to impair gonadal development [11, 12], alter sex phenotypes [12, 13] and to feminise and/or masculinise individuals [1, 13, 14].

The zebrafish (*Danio rerio*) is recommended as a test species in many existing standard ecotoxicological guidelines and it is probably the most studied fish in developmental biology [2, 8]. Zebrafish is considered to be an undifferentiated gonochoristic species, with both sexes passing through an ovary-like or juvenile-ovary stage, before differentiation into both phenotypic mature sexes. In males, this includes a type of juvenile hemaphroditism at around 25 days post hatching [15, 16]. Sexual determination in zebrafish is complex. Although it is described to be polygenic without distinguishable sex chromosomes in domesticated laboratory strains [17, 18], natural populations possess a WZ/ZZ sex chromosomes system [18]. On top of that, sex determination is secondarily influenced by environment [7, 10, 19]. Exposure to different chemical compounds considered EDCs at relatively low concentrations can cause sex reversal and disturbances in gonad development and reproductive output [12, 19].

Traditional biomarkers of fish exposure to (xeno)hormones include alterations in gross morphology and sex characteristics, changes in gonadosomatic index and in plasma or liver vitellogenin levels, and presence of gonad histopathological alterations [3, 4, 7, 20]. One of the most notable effects identified in fish inhabiting polluted sites is the onset of intersex condition, where male testis develops oocytes within the spermatogenic cysts [3, 20]. Exposure to reproductive EDCs also alters the normal expression pattern of genes involved in sex differentiation. Some of the genes whose transcription levels are dimorphic in relation to sex phenotype and whose transcription is altered after exposure to EDCSs are the anti-Müllerian hormone (*amh*), doublesex and mab-3 related transcription factor 1 (*drmt1*) and gonadal aromatase (*cyp19a1a*). It is well known that exposure to androgens down-regulates *cyp19a1a* expression [21] during ovarian development. Thus, *cyp19a1a* which encodes the aromatase responsible for the biosynthesis of estradiol from testosterone in fish gonads exhibits sexually dimorphic expression [1]. In contrast, male marker genes, such as *amh* and *dmrt1*, are down-regulated after estrogen exposure leading to downstream feminizing effects [22, 23]. In an environmental context of xenoestrogen exposure in which intersex males appear [3, 6, 20] it remains to be elucidated whether transcription alteration of such genes reveals estrogen/xenobiotic exposure or oocyte formation in testis [24]. It follows, that generating biomarkers of

oocyte differentiation could be important for the early identification of intersex condition in environmental monitoring programs.

In addition to the mentioned genes, the general transcription factor IIIA (*gtf3a*), controlling the transcription of 5S rRNA in eukaryotes [25], has been shown to be up-regulated in testis of intersex male thicklip grey mullets (*Chelon labrosus*) from polluted harbours in the Bay of Biscay [6, 20, 26,]. These mullets also showed elevated vitellogenin transcript and protein levels in liver and plasma and up-regulation of *cyp19a1b* in brain [4]. 5S rRNA is strongly expressed and accumulated in teleost oocytes working as a very efficient molecular marker of sex in teleost fish [20, 26]. Transcription of 5S rRNA is especially strong during early oogenesis in previtellogenic oocytes and in this way, its transcription levels can be used to infer molecularly female gametogenic stage in teleost fish [20, 27]. Such cytosolic accumulation of 5S rRNA facilitates that in case of successful fertilisation ribosomes will be quickly assembled to assist protein synthesis during early embryo development [25]. *gtf3a* transcription levels consequently are higher in ovaries than in testes, as it has been certified in all teleost fishes tested to date [27]. This occurs because Gtf3a does not only act as a transcription factor for the activation of RNA polymerase III, but it also acts as a 5S rRNA binding protein for its stockpiling in the cytosol [26, 27]. In *Xenopus laevis*, a single *gtf3a* gene codes for two different transcripts corresponding to an oocyte and a somatic form of the protein, synthesised through differential promoter usage [28, 29]. In fish, a single *gtf3a* gene was first characterised in the catfish *Ictalurus punctatus*, where it was shown to code for a protein associated to 5S rRNA in the oocytes, but possibly not involved in transcriptional regulation of 5S rDNA [30].

Thus, the objectives of the present study were to elucidate the nature of the *gtf3a* sexually dimorphic transcription in fish gonads. Not knowing whether fish present two differentially expressed *gtf3a* transcripts, one specifically expressed in ovaries and another one in somatic tissues, as it is the case in anuran frogs, we intended to characterise the *gtf3a* repertoire in teleost genomes. Further, we wanted to elucidate whether transcription of *gtf3a* in ovaries (and in intersex testis) is a consequence of oestrogen exposure or of oocyte differentiation. In order to pursue such objectives, we selected the laboratory model species *Danio rerio*, whose genome is fully sequenced, and studied the pattern of *gtf3a* transcription together with that of other well-characterised sex differentiation genes under exposure to 17β-estradiol and 17α-methyltestosterone from fertilisation to 60 days post-fertilisation (dpf).

## Results

### Synteny analysis of the general transcription factor 3A gene (gtf3a)

To elucidate the *gtf3a* evolutionary history, we compared the adjacent genomic regions of *gtf3a* in all curated vertebrate genomes incorporated in the Ensembl gene browser 96. In non-teleost genomes a single *gtf3a* was identified (Figs 1 and 2). *gtf3a* gene was observed to be duplicated (*gtf3aa* and *gtff3ab*) in all *Osteophysi* and *Euteleostei* teleost genomes present in Ensembl 96. Instead, only one gene, identified as *gtf3ab*, is present in the Asian bony tongue and in *P. kingsleyae* (*Osteoglossomorpha*). The same can be said about the other *Ostoeglossomorpha* whose genome has been sequenced, the pirarucu *Arapaima gigas* [31]. In *Osteophysi* and *Euteleostei* both paralogs appeared always in different chromosomes or scaffolds (Fig 2). For instance, in the case of zebrafish *gtf3aa* was present in chromosome 5, while *gtf3ab* was present in chromosome 24. Phylogenetic analyses showed that the Gtf3ab protein sequences clustered together, separated from the cluster formed by the protein sequences of *gtf3aa* genes and the non-teleost *gtf3a*s (Fig 1). Synteny analysis, revealed that teleost *gtf3ab* neighbouring genes coincided greatly with those surrounding the *gtf3a* genes in all vertebrate species studied (Fig 2). In turn, *gtf3aa* in teleost genomes did not conserve any of the non-

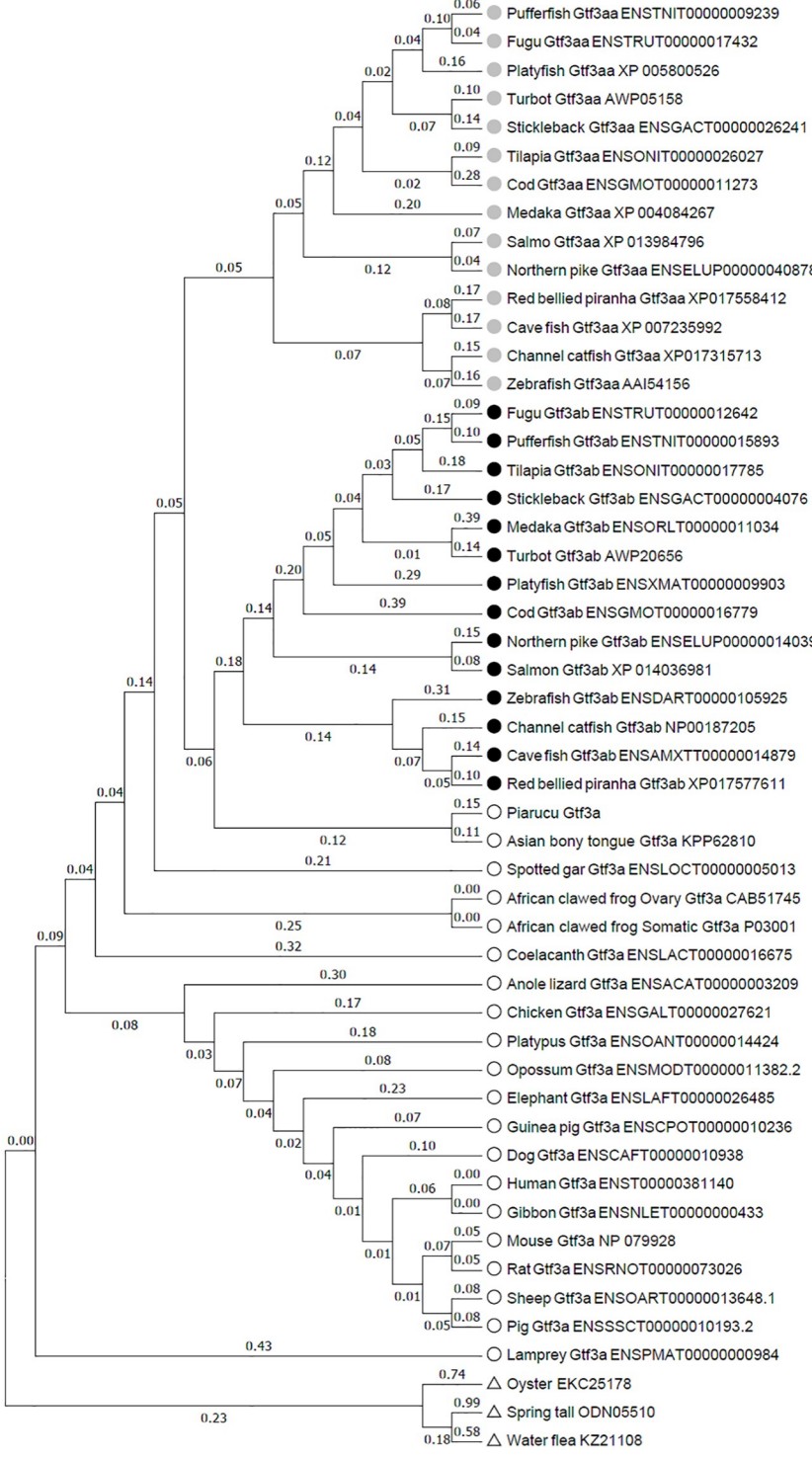

**Fig 1. Phylogenetic tree showing the relationship between different animal deduced Gtf3a protein sequences.** Sequences were aligned with MEGA7 using MUSCLE (v3.8.31) conFig d for highest accuracy. The evolutionary history was inferred by using the Maximum Likelihood method based on the JTT matrix-based model. The tree with the highest log likelihood (-8390.8717) is shown. Initial tree(s) for the heuristic search were obtained automatically by applying Neighbor-Join and BioNJ algorithms to a matrix of pairwise distances estimated using a JTT model, and then selecting the topology with superior log likelihood value. The tree is drawn to scale, with branch lengths measured in the number of substitutions per site (next to the branches). The analysis involved 51 protein sequences. All positions containing gaps and missing data were eliminated. There were a total of 157 positions in the final dataset. All Gtfab proteins are indicated with black circles, with grey circles for Gtfaa, and white circles for Gtfa in non teleosts and in *Ostoglossomorpha*. Triangles mark proteins coded by invertebrate Gtf2a orthologs used as outgroups.

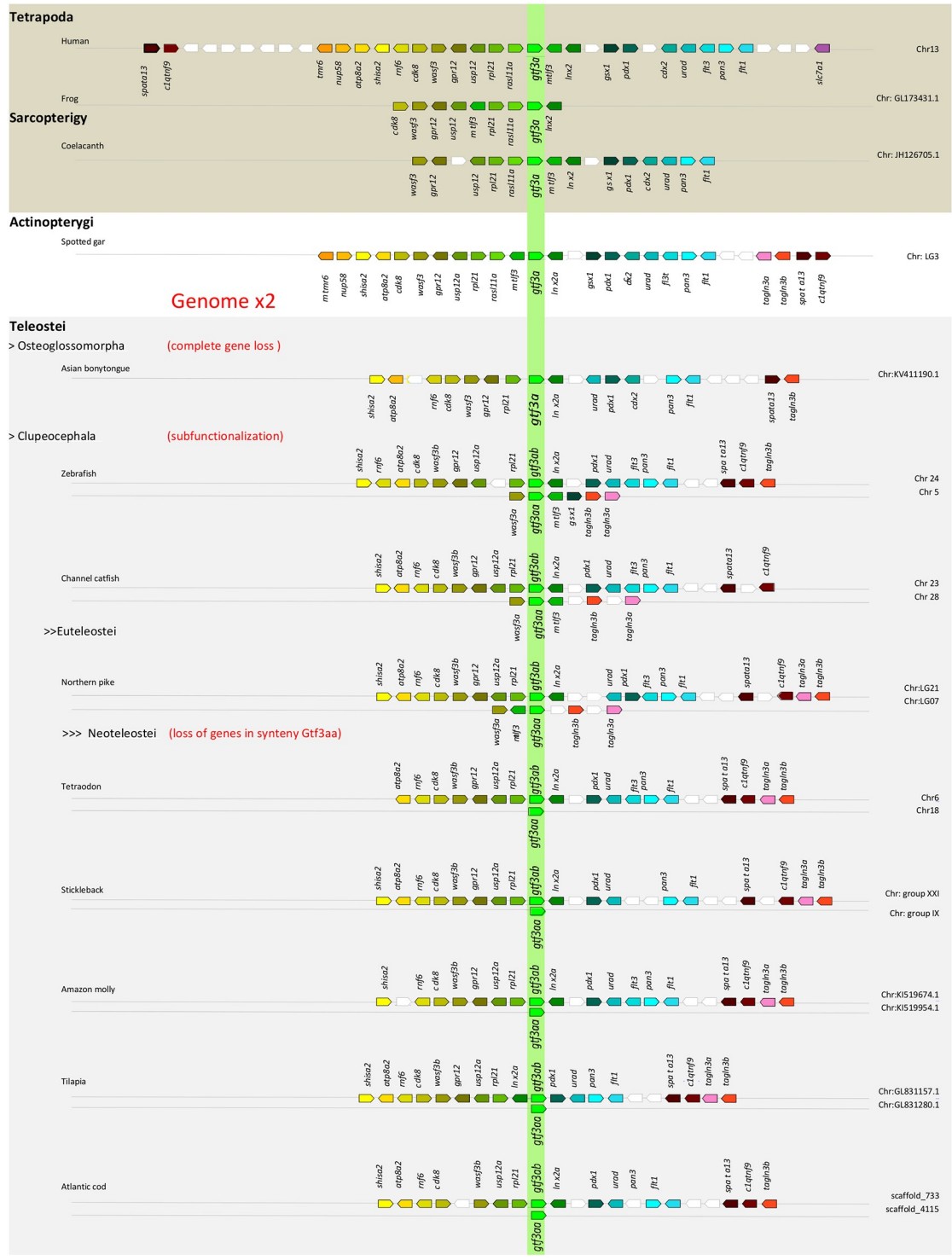

**Fig 2. Conserved genomic synteny of vertebrate *gtf3a* genes.** Genomic synteny maps comparing localisation *gtf3a* and neighbouring genes in the genomes of tetrapods (human and frog), a basal sarcopterygian (coelacanth), a non-teleost actinopterygian (spotted gar) and different teleost species (Asian bonytongue, zebrafish, channel catfish, northern pike, tetraodon, stickleback, amazon molly, tilapia and Atlantic cod). In teleosts, with the exception of Asian bonytongue, two *gtf3a* paralogs exits (*gtf3aa* and *gtf3ab*) and the neighbouring genes are shown in comparison to the genes surrounding human *gtf3a*. Orthologs for each gene are represented with the same colour and displayed in the order and direction in which they are placed in each chromosome or scaffold as indicated in the right side of the Fig (detailed genomic locations for each gene are given in S1 Table).

teleostean *gtf3a* neighbouring genes, with the exception of *Osteophysi* and Northern pike (*Esocifomes*). In these cases, *wasf3*, *mtif3*, *gsx1* (in the case of zebrafish), *abhd10a*, *tagln3a and shisa2b* were present in synteny (and also duplicated) close to *gtf3aa*. Also in the genomes of salmonids, sister group of *Esociformes*. Special is the case among all teleosts, of the Orange clown fish where besides *gtf3ab* (ENSAPEG00000019970), two *gtf3aa* genes (ENSAPEG0000 0007319 and ENSAPEG00 000007343) are repeated in tandem in the same frame in chromosome number 3. For a complete analysis of the synteny of *gtf3aa* and *gtf3ab* genes in fish genomes see S5 Fig.

In zebrafish genome observed paralogs, *gtf3aa* (ENSDARG00000030267) in chromosome 5 and *gtf3ab* (ENSDARG00000071583) in chromosome 24, code for two different proteins with a deduced protein size of 367 aa and estimated molecular weight of 42,6 kDa in the first case, and 318 aa with 37.15 kDa for *gf3ab*. The deduced amino acid sequence of all teleostean Gtf3ab proteins allows to observe the conserved initiating sequence MGER(K) (S3 Table). On the other hand, the KRSLAS domain behind the last Zn finger C2H2-type domain in the protein (S3 Table), is only present in teleostean Gtf3aas but not in Gtf3abs.

## Organ specific transcription of piscine *gtf3a* paralog genes and dynamics of *gtf3aa* and *gtf3ab* transcription during zebrafish early embryo development

Transcription levels of both *gtf3a* paralog genes were studied in brain, gonads and muscle of adult zebrafish (Fig 3A). While *gtf3aa* was strongly transcribed in all the tissues studied, *gtf3ab* showed an ovary specific transcription, with a hint of transcription in the testis (agarose gel analysis of End-Point RT-PCR results). The same transcription pattern was observed in gill, eye, ovary and testis (S1 Fig) of tilapia. Results were verified by *gtf3ab* qPCR analysis in muscle, testis and ovaries of zebrafish (Fig 3B). Brain transcription was not studied by qPCR due to the inexistent transcription signal in the electrophoresis. *gtf3ab* transcript levels were high in ovaries and nearly non-existent in muscle and testis, confirming the specific transcription of *gtf3ab* in ovaries. *gtf3aa* and *gtf3ab* transcription levels were analysed during early embryonic development, and transcript levels were detected at 2 hpf (64-cell blastula), 8 hpf (gastrula period) and 30 hpf (Prim-16 period) for both genes (Fig 3C). The transcription levels of *gtf3aa* suffered a reduction at 8 hpf increasing again at 30 hpf. In contrast, *gtf3ab* transcript levels decreased after 2 hpf and were not observable after 8 hpf. A similar pattern of gene transcription in relation to ovary vs other tissues and in relation to early embryogenesis is suggested by the RNA-Seq data present in Ensembl for zebrafish (S3 and S4 Figs). The RNA-Seq data studied in pirarucu shows that in this species containing one *gtf3a* gene, this is strongly expressed in the ovary (S4 Table).

## Gonad development in zebrafish exposed to estrogenic hormones

Histological analysis was performed in 10 individuals from each experimental group at 26 and 61 dpf (Fig 4). No gonad was present in any of 26 dpf individuals with the exception of an ovary identified in one individual in the ET control group. After 61 dpf differentiated gonads were observed in all individuals with 60% of individuals showing ovaries and 40% showing testes in the case of the ethanol control group. 100% of the individuals were females, displaying well differentiated ovaries, in the 17β-estradiol (E) treatment group. 100% were males in the 17α-methyltestosterone (MT) treatment group. There was no difference regarding gonad development between control and treated groups. Ovaries always showed previtellogenic oocytes and testis were always in early-mid spermatogenesis. After a year in clean water both E and MT groups were constituted by monosex populations.

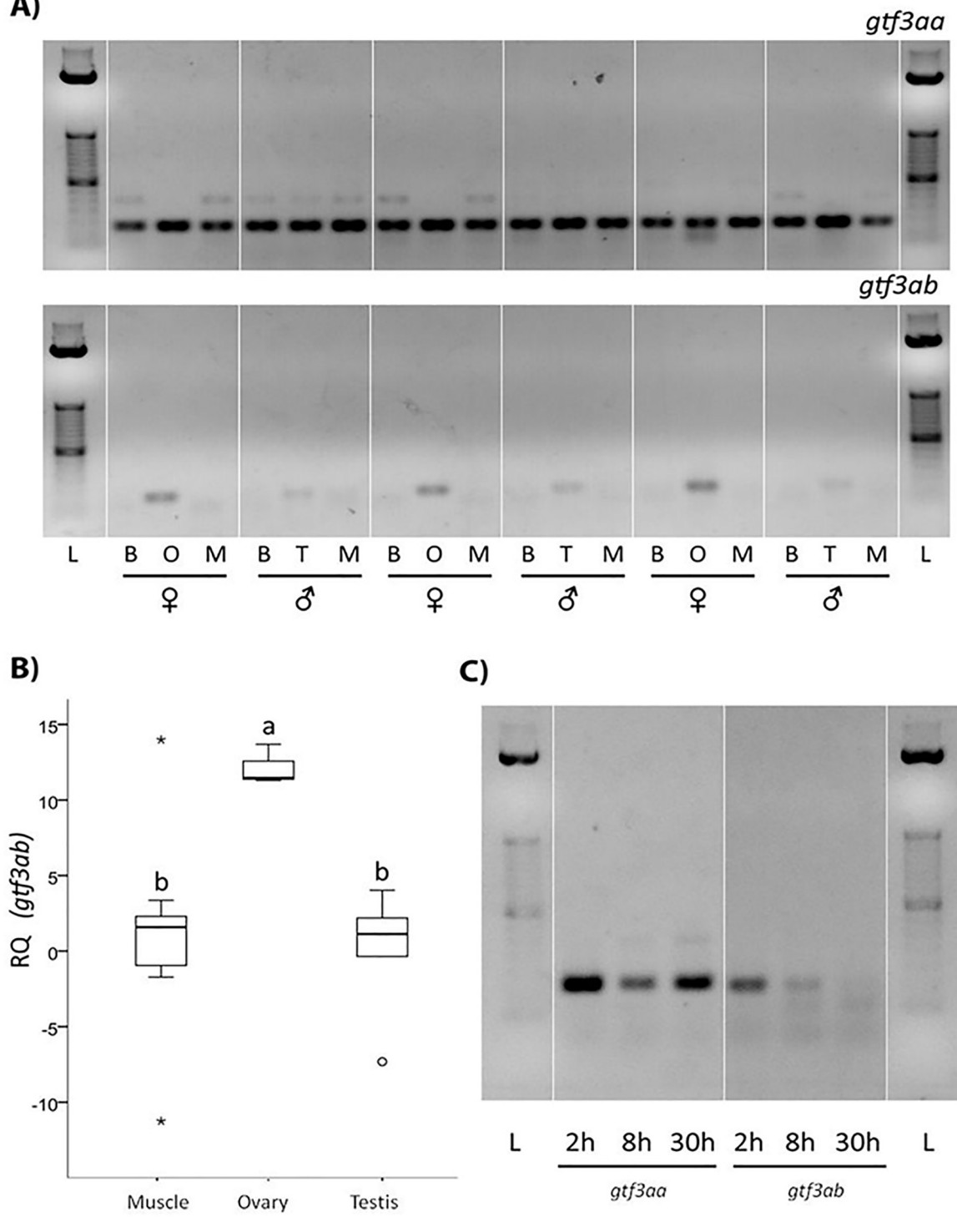

**Fig 3. Transcription levels of *gtf3aa* and *gtf3ab* in different tissues and developmental stages of zebrafish.** A) Agarose gel electrophoresis showing presence vs absence of *gtf3aa* and *gtf3ab transcript levels* after End-Point RT-PCR in organs of three male and three female individuals. Brain (B), ovary (O), testis (T) and muscle (M). Amplified fragments of 100 (*gtf3aa*) and 114 (*gtf3ab*) nucleotides were observed in each case. L = Standard 50 bp (Invitrogen). B) Transcription levels of *gtf3ab* in different tissues of adult zebrafish through qPCR. Box plots represent the data within the 25th and 75th percentiles, with the median indicated by a line, and top and bottom whiskers indicating the minimum and maximum values. Different letters indicate significant differences between means (Mann Whitney, p<0.05). Number of samples, 6 for ovary, 3 for muscle and 4 for testes. C) Agarose gel electrophoresis afterEnd-Point RT-PCR amplification showing transcription levels of *gtf3aa* and *gtf3ab* in zebrafish embryos 2, 8 and 30 hpf. Amplified fragments were around 100 nucleotides in size. L = Standard 50 bp (Invitrogen).

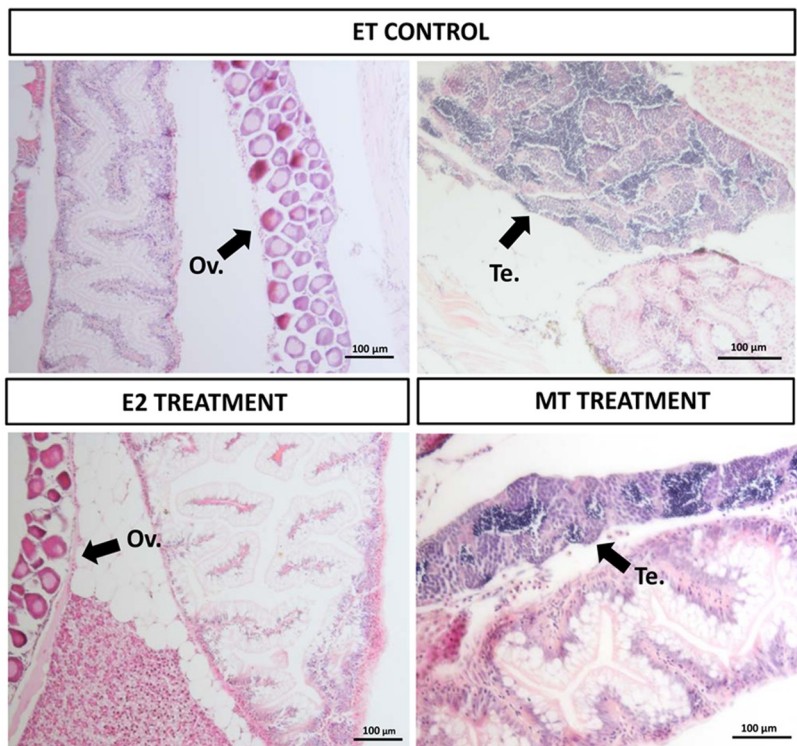

**Fig 4. Histological analysis of hormone treated zebrafish at day 61.** Micrographs (scale bars = 100 μm) are representative of 10 individuals analysed for each experimental group. ET: ethanol control group (6 females with well formed ovaries and 4 males with testis), E. 17β-estradiol treatment (100% of the individuals with ovary) and MT: 17α-methyltestosterone treatment (100% of the individuals with testis). Black arrows mark the presence of well differentiated gonad (ovaries in the two micrographs in the left and testes in the right).

## Gene transcription profiles in feminised and masculinised zebrafish

The results of all the individuals from both replicate tanks per experimental group were used together (n = 12) as no differences were observed between tanks in growth (S2 Table) (Mann-Whitney, p>0.05) or in gene transcription levels. Target gene transcription levels in larvae at day 26 showed no differences between feminised and masculinised groups (Fig 5). Only *gtf3ab* showed higher transcription levels in the ET control group than in the E exposed group. This difference existed also with the MT group although it was not significant. Differences in transcription levels between hormonally feminised and masculinised fish were observed at day 61, with higher transcription levels of *gtf3ab* in the E exposure group. These analyses were performed on RNA extracted from whole organisms without dissection of organs. Transcription levels of *amh* and *dmrt1* were higher in the MT exposure group than in the E group. *cyp19a1a* transcription levels were maintained constant in all groups, as it was the case also for *gtf3aa* and *actb*. The control group showed values in between the two treatment groups at 61 dpf. *dmrt1* was up-regulated in the MT group at 61 dpf in comparison to 26 dpf. In contrast, *gtf3ab* was up-regulated in the E group at day 61 *vs* day 26 (Fig 5).

The high variability observed for some of the genes within the ET61 control group should be due to the the presence of both female and male individuals. Attending to the bimodal levels of transcription of *gtf3ab*, high levels indicating females (S2 Fig), we distinguished the female and male individuals within ET control group. ET females identified in this mode showed significantly higher *cyp19a1a* transcription levels than ET males, with a down-regulation

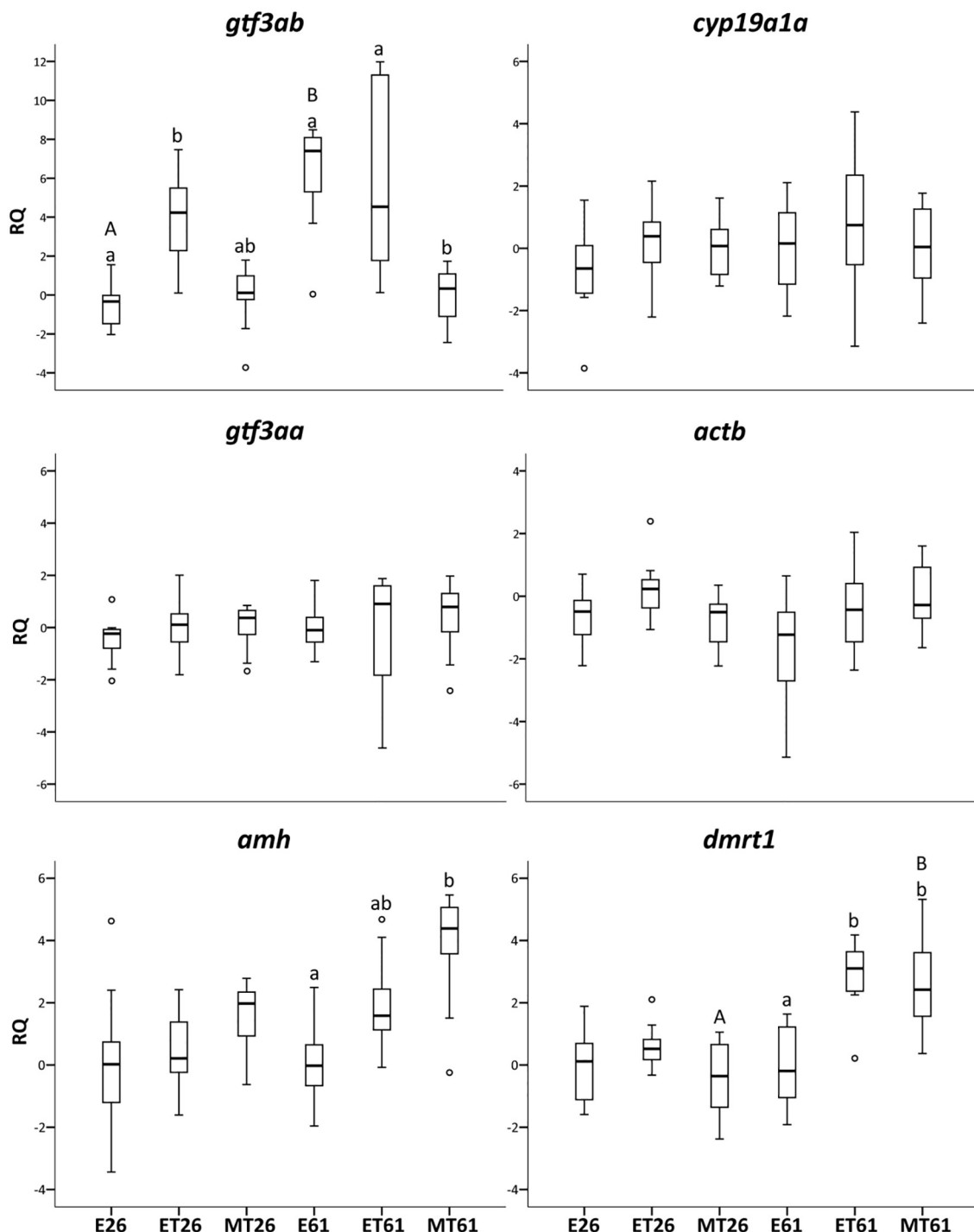

**Fig 5. Transcription levels of sex related genes in zebrafish exposed to hormones for 26 and 61 days.** Fish exposed to 17β-estradiol (E) and 17α-methyltestosterone (MT) for 26 and 61 days (E26, E61 and MT26, MT61). Ethanol control group at 26 and 61 days (ET26 and ET61). Box plots represent the data within the 25th and 75th percentiles, with the median indicated by a line, and top and bottom whiskers indicating the minimum and maximum values (n = 12 individuals per experimental group). Different lower case letters indicate significant differences between groups within each sampling day and different capital letters indicate significant differences within each exposure group either comparing days 26 and 61 (Kruskal-Wallis, p<0.05).

associated to E and MT treatments (S2 Fig). ET group females and males showed no differences in *amh* and *dmrt1* transcription levels, but E treatment resulted in a down-regulation of *dmrt1* when compared to control males and females and a down-regulation of *amh* when compared to control males (S2 Fig).

## Discussion

Phylogenetic and synteny analysis has demonstrated the presence of a single *gtf3a* in coelacanth, spotted gar, *Osteoglossomorpha* and in all tetrapods. Actinopterygians, diverged before the teleost-specific third whole genome duplication (3R), present a single *gtf3a* gene while teleost fish genomes present two *gtf3a* genes (*gtf3aa* and *gtf3ab*), with the exception of early teleosts *Osteoglossomorpha*. In zebrafish *gtf3aa* would code for a protein of 367 aa and and estimated molecular weight of 42,6 kDa, this values being i 318 aa and 37.15 kDa for the case of *gf3ab*. This is very close to the two protein sequences obtained through alternative promoter usage of the unique *gtf3a* gene present in the *X. laevis* genome [29, 30]. The smallest transcript gives rise to a 38 kDa protein that is observed in all studied frog tissues, while the biggest one with 40 kDa is produced only in the ovary [28]. We show hereby that the deduced amino acid sequence of all teleostean Gtf3ab proteins begins with the sequence MGER(K) typical of the oocyte form of the protein in all frog species [30]. On the other hand, the KRSLAS domain behind the last Zn finger C2H2-type domain in the protein, required for Gtf3a-dependant 5S gene transcription [30], is only present in teleostean Gtf3aas. This could suggest subfunctionalisation of *gtf3ab* in teleosts, with an oocyte specific function for Gtf3ab in 5S rRNA binding and stockpiling without a role as transcription factor. This function would be retained only by Gtf3aa in all cell types.

Synteny analysis reveals that the region neighbouring *gtf3ab* in teleosts, very conserved in all Ensembl 96 teleost genomes, is most similar to that of other vertebrates, including Asian bonytongue, spotted gar, coelacanth, mammals, birds, reptiles and amphibians, something that does not occur with *gtf3aa*. We have shown that the *gtf3aa* neighbouring region in all *Osteophysi* genomes available in Ensembl 96 (zebrafish, Mexican tetra, cave fish and channel catfish) and further in Northen pike (*Esox lucius*), conserve some neighbouring genes that clearly point to the teleost specific genome duplication event (TGD or 3R) as the cause of the appearance of both paralog genes. Similar results have been obtained through synteny analysis of other genes, for instance the *hox* cluster in the *Elopomorpha*, *Osteoglossomorpha* or *Clupeocephala* fish [31, 32, 33]. The existence of duplicated genes in relation to reproductive endocrinology and sex differentiation control has been extensively documented in many fish species [34]. For *gtf3aa* and *gatf3ab* in Neoteleostei, a complete loss of duplicated genes might have occurred after the TGD that finally has resulted in the maintenance of only a duplicated paralog copy of *gtf3aa*. The fact that no *gtf3aa* orthologs are found in *Osteoglossomorpha*, *Paramormyrops kingsleyae* and *Scleropages formosus* and *Arapaima gigas* [31] in the base of teleosts, would suggest that after the TGD the duplicated genes, including *gtf3aa*, were lost. In this sense, call the attention on the fact that the *gtf3a* gene present in these *Osteoglossomorpha* fish produces a protein that clusters closer to other teleost Gtf3ab-s (Fig 1) without showing its characteristic amino-terminal end sequence. Further conclusions will have to await to more complete genome sequence assemblages of basal teleosts.

The possibility of a gene specifically transcribed in ovaries, *gtf3ab*, and another gene, *gtf3aa*, transcribed in somatic tissues and testis has been confirmed in this study. End-Point RT-PCR results demonstrated that both paralogs are differentially transcribed in different tissues of zebrafish and tilapia. In both species *gtf3ab* was highly transcribed in ovaries but not in other tissues. In contrast, *gtf3aa* was transcribed in all the studied tissues, including ovaries. This has

been described in frogs, but in this case with two proteins produced through alternative promoter usage of one single gene [28, 29]. Previous studies performed in our laboratory comparing gonads of thicklip grey mullets along a complete reproductive cycle demonstrated high transcription levels of *gtf3a* (identity analysis in the light of the present results reveals that this sequence, JN257141, in fact belongs to mullet *gtf3ab*) in ovaries and not in testis, all along the cycle [26]. Moreover, studies performed in megrim (*Lepidorhombus whiffiagonis*) and European anchovy (*Engraulis encrasicolus*) have demonstrated that, *gtf3ab* is not only differentially transcribed in ovaries comparing with testis, but also in ovaries at different developmental stages. *gtf3ab* transcription is at its highest early in oogenesis to decrease during later stages, in association with a decrease in 5S rRNA transcription and an activation of 18S and 28S rRNA production during secondary oocyte growth [27]. The linkage of transcription of *gtf3ab* to ovarian tissue is also apparent in the RNA-Seq data available for zebrafish in Ensembl. In the pirarucu, where only one *gtf3a* gene exists, RNA-Seq analysis reveals highest trancription levels of this gene also in ovaries.

Traditional RT-PCR analyses during the first hours of zebrafish embryo development demonstrate the maternal oocyte origin of *gtf3ab* transcripts as corroborated by the RNA-Seq data available in Ensembl. At 64-cell stage (2 hpf), when zygotic genome is not transcribed in zebrafish yet, *gtf3ab* transcript levels were high, decreasing as embryogenesis proceeded towards Prim-16 stage (30 hpf). After fertilisation, maternal mRNA factors support early embryonic development until activation of zygotic transcription [35]. The initiation of zygotic transcription occurs during the "maternal-embryo transition" (MET). In fish, MET occurs at the mid-blastula stage and it is also known as "mid-blastula transition" (MBT). In zebrafish MBT is well characterised and takes place at 512-cell stage after 2.75 hpf. At this time, and until 50% epiboly stage (5.25 hpf), a major transition in gene regulation and transcriptional activity takes place [36]. The disappearance of *gtf3ab* transcripts after 2 hpf reveals the maternal (oocytic) origin of *gtf3ab*, which is linked to ovarian 5S rRNA production in oocytes, and thus to ribosome formation that would ensure fast protein production during early embryonic developmental.

In contrast, *gtf3aa* transcript levels decreased slightly from the 2 hpf embryo to the gastrula stage (8 hpf). Then, and as a consequence of MET at 30 hpf, *gtf3aa* transcript levels increased to levels observed at 2 hpf. At this stage (18 hours before hatching), the 2.5 mm embryo is undergoing the last organogenesis processes [35]. Our research has shown that no gonad has been formed yet at this stage and until ovarian tissues are formed, no zygotic *gtf3ab* transcription can be observed.

Gonad differentiation in zebrafish occurs between 25 and 45 dpf and is completed after 60 dpf [15]. In our case, no visible gonad was observed in treated and non-treated individuals after 26 dpf. This absence of gonad could have been caused by the water temperature of 24˚C during the experiment. Zebrafish optimal growth temperature ranges from 26 to 32˚C and it has been previously described that lower temperatures could cause a delay in general growth of zebrafish, as well as in gonad differentiation and maturation [37]. In addition, exposure to oestrogens has been shown to reduce fish growth in a concentration-dependent way [2, 38]. Androgens used for 96 hours to obtain a medaka male monosex populations [5] increased growth rates. In the present study no significant differences in growth were observed among individuals in the control, the E or the MT groups.

No sex reversal was observed in both generated monosex zebrafish groups after 1 year in clean water. Baumann and colleagues [39] reported that the irreversibility of the androgenic effects on sexual development in zebrafish is a consequence of loss of primordial germ cells during early testis development, making the later development of ovaries impossible. In contrast, it has been described that feminised genetic males could develop testis after withdrawal

of estrogenic compounds [11, 40]. This reversible effect could be dependent on exposure timing and concentration and on duration of the treatment [41]. Exposure during gonad differentiation at early life stages, as in our experiment, would make the process irreversible.

Many studies have demonstrated that fish exposure to hormones during sex differentiation disrupts the normal expression of genes involved in gonadogenesis [1, 21]. In the present study, where the whole body transcription levels were analysed without dissection of specific organs, no changes in the transcription levels of studied genes were observed after 26 days of exposure, with the exception of *gtf3ab*, down-regulated in both hormone treatment groups. This lack of transcriptional responses seems to be linked to the fact that in the present experiment zebrafish did not present developed gonads at 26 dpf. In normal conditions, zebrafish gonad differentiation starts with a juvenile ovary phase from 20 dpf to around 30 dpf [15, 16]. It is possible that some individuals from the ET control group might have initiated gonad development as juvenile females at 26 dpf, and this would be reflected in the higher transcription levels of ovary specific *gtf3ab* in comparison to hormonal treatment groups.

After 61 days of exposure, E feminised group showed up-regulated *gtf3ab* in comparison to E at day 26 and MT at day 61, MT group in turn showing down-regulation in respect to the control group. In the control group, with individuals of both sexes being present, a strong variability in *gtf3ab* transcription levels was observed resulting from a bimodal (male *vs* female) transcription pattern.

In contrast, E exposure showed a suppressive effect on *amh* and *dmrt1* transcription levels, which are associated with male sex differentiation [22, 42]. Schulz and colleagues [23] observed that 5 ng/L ethinyl estradiol exposure during zebrafish early life stages suppressed both *amh* and *dmrt1* expression and caused an inhibition of male gonad development. The MT masculinised group showed up-regulation of both genes. Transcription levels for these two male marker genes were recorded in the ET group, due to the presence of both female and male individuals in this group. Male-specific differentiation in mammals includes activation of *amh* expression as the first factor secreted by differentiated Sertoli cells in the testis and leading to the regression of Müllerian ducts. Despite de absence of Müllerian ducts in teleosts, they present an *amh* ortholog [22] that is mainly expressed in males during sex differentiation, suggesting that it has a function during testis differentiation and spermatogonial proliferation [6, 43, 44]. *Dmrt1* is expressed in Sertoli cells after testicular differentiation for spermatogonial proliferation [42]. In zebrafish, *dmrt1* expression is up-regulated during the early testicular differentiation, but it is also observed in ovarian developing germ cells. *dmrt1* induces male phenotypic development via a down-regulation of aromatase shifting the steroidogenic pathway towards androgen production [45].

In this sense, *cyp19a*, encodes the enzyme aromatase, which is responsible for catalyzing the aromatisation of androgens to oestrogens, being a key gene in ovarian differentiation in teleosts [46]. The role of *cyp19a1* genes in ovarian differentiation has been demonstrated in several studies that have treated fish with aromatase inhibitors, resulting in the suppression of oestrogen biosynthesis and induction of sex-reversal of genetic females to phenotypic males [21, 47, 48]. It has been shown that the *cyp19a1a* promoter region does not display an oestrogen response element in teleosts and it is not transcriptionally activated by oestrogens as observed in the present study, in contrast to what occurs with *cyp19a1b* in the brain [4, 6, 49]. Alternatively, phenotypic masculinisation occurs when fish in early life stages have been treated with androgens is accompanied by a down-regulation of *cyp19a1a* [21, 48]. However, since aromatase converts androgens to estrogens, exposure to aromatizable androgens (i.e. MT) may induce both masculinizing and feminizing effects [49, 50] and this could explain the lack *cyp19a1a* down-regulation in the MT group in comparison to the E group seen hereby. The present studies are showing transcription levels using RNA extracted from whole individuals

not only gonads, pointing out also to the added value of *gtf3ab* as a marker of oocyte differentiation in fish.

No differences were observed in *gtf3aa* transcription levels, as it occurred with *actb*, not being affected by sex or hormone exposure. Several studies use *actb* as a reference gene in PCR analyses due to its constant transcription throughout tissues and experimental conditions in fish [51]. *actb* transcription levels have been reported to change in fish gonads after E exposure [52]. This is why, and in spite of the lack of variability of *actb* transcription in the present study involving whole body RNA, we suggest when working with fish gonads to refer qPCR results to the amount of cDNA amplified per sample [6, 20].

All together, two *gtf3a* paralog genes are present in teleost genomes due to the TGD event. As a consequence of sub-functionalisation of the new gene products, while *gtf3ab* displays ovarian specific transcription associated with ovarian 5S rRNA stockpiling, *gtf3aa* has been maintained as a gene for the transcriptional regulation of 5S rRNA in all somatic tissues, but also in testis and ovary. Exposure to hormones (E and MT) affecting sex differentiation in zebrafish demonstrate that transcription of *gtf3ab* is a consequence of oogenesis and a marker of oocyte differentiation, not a marker of oestrogen or androgen exposure *per se* (differences observed at day 61, not at day 26). This circumstance, that needs to be proved true in other fish species, has important consequences for pollution monitoring programmes. Oocyte specific transcription of *gtf3ab* has already been described in intersex testis of *C. labrosus* from polluted Basque estuaries [3,6, 26], further defining *gtf3ab* transcripts as specific molecular markers of intersex condition and oocyte production in testes, and not mere xenoestrogen exposure.

## Materials and methods

### Synteny analysis of the general transcription factor 3A gene (gtf3a)

The Emsembl genome repository was [53] searched looking for vertebrate *gtf3a* sequences ortholog to the known *Chelon labrosus* (JN257141) and *Xenopus laevis* (BC129561) sequences. When such sequences were found the possible presence of paralog sequences in each of the genomes was checked. Then, a synteny analysis was carried out using Spotted gar *Lepisosteus oculatus gtf3a* (Ensembl ENST00000381140) as template and comparing all the flanking genes for each of the identified *gtf3a* genes using the Genomicus v96.01 web-tools [54]. Phylogenetic and molecular evolutionary analyses were conducted using MEGA version 7 [55]. The protein sequences were coded by *gtf3a*, *gtf3aa* and *gtf3ab* genes in fish (teleosts and the non-teleost actinoptygian fish *L. oculatus* and basal sarcopterygian fish *Latimeria chalumnae*) and selected tetrapods. Oocyte and somatic Gtf3a were included in the case of Xenopus, and invertebrate *gtf3a* orthologs as outgroups.

### Zebrafish breeding and hormonal treatment to obtain two monosex stocks

**Chemicals.** In order to obtain monosex female and male populations, zebrafish were exposed from fertilisation to 61 dpf to 100 ng/L of 17β-estradiol (E) and 17α-methyltestosterone (MT) taking into account the procedures described by Andersen et al. [56]. Hormones (purity >98%) were obtained from Sigma-Aldrich (St. Louis, MO, USA). E and MT stock solutions were prepared fresh weekly dissolving them in absolute ethanol at 1 g/L (stock solution). Then, 2 mL of stock solutions were daily diluted in 20 L water to obtain a final E or MT nominal concentration of 100 ng/L and 0.01% ethanol in the total volume.

**Zebrafish egg production and exposure to hormones.** Adult zebrafish (*Danio rerio*, wild type AB Tübingen) were maintained at a water temperature of 24˚C with a 14 h light/10 h dark cycle in 100 L tanks with mechanical and biological filters following standard protocols

as described in Vicario-Parés et al., 2014 [57]. The fish were fed with Vipagran Baby (Sera, Heinsberg, Germany) and artemia nauplii (*Artemia salina*) twice per day. 23 breeding couples were selected and placed separated by sex through a barrier in a single tank. The day prior to the beginning of the exposure experiment, females and males were independently coupled in breeding traps separated by a barrier. Before turning on the light in the following morning, the barrier was removed. 18 couples reproduced and the newly fertilised eggs were collected selecting viable ones under a stereoscopic microscope (Nikon smz800, Kanagawa, Japan).

Obtained embryos (n = 830 initial amount) were separated in groups of 40 to 60 individuals and placed in glass Petri dishes, to obtain three Petri dishes per experimental group. The Petri dishes were filled with 50 mL of corresponding experimental solution: water (water control), water with 0.01% ethanol (ethanol control, ET), and hormone solution (100 ng/L E or MT depending on the experimental group). Once a day half of the volume was replaced with fresh solution. Additionally, embryos were examined daily until 5 dpf to detect and discard any embryo with malformations or dead. Criteria for normal zebrafish embryo development morphology were based on Kimmel et al. [58].

After 5 dpf 100 larvae were transferred to 10 L tanks (filled with 8 L control or spiked water) and exposed in constant drip flux until day 61. The experiment was carried out in duplicated tanks for each experimental condition. The exposure regime was 10 L/day replaced with new solution (S1C and S1D Fig), and pH, conductivity, ammonia, nitrates and nitrites were weekly measured using commercial Sera tests (Hersteller, Germany) for the chemical analysis. The juveniles received food *ad libitum* twice a day with corresponding fish food and artemia nauplii. Waste produced by fish was carefully sucked away and lost volume immediately replaced. After 26 and 61 dpf, 22 individuals from each experimental condition were euthanised by an overdose of MS-222 (tricaine methane-sulfonate, Sigma-Aldrich). 12 whole individuals (6 from each experimental group replicate) without dissection were independently embedded in RNA later (Ambion, Life Technologies, Carslbard, USA) and frozen at -80˚C for molecular analysis. 61 day juveniles were frozen after decapitation. The remaining 10 individuals were fixed in 4% neutral buffered formalin (NBF) and stored at 4˚C for 24 hours before paraffin embedding for histological analysis. All individuals were measured upon sampling. After 61 dpf remaining fish were kept in clean water for 1 year and then sexed. All along the experiment, no appreciable mortality was observed in any of the groups or replicate tanks.

### Sampling of adult zebrafish and tilapia individuals

Organs were dissected from three female and three male adult zebrafish (UB Tubingen) from our own stock. Adult tilapias (*Oreochromis niloticus*), five females and four males, were obtained from BREEN, Ltd., (NER group, Hondarribia, Spain). Fish were anaesthetised in a saturated ethyl 4-aminobenzoate (Fluka, Steinheim, Germany) water bath. The methods in this section were carried out in accordance with the approved guidelines. Each fish was sacrificed by decapitation and gonad, liver, muscle, eye and brain were dissected. A portion of each tissue was embedded in RNA later (Ambion, Life Technologies), frozen in liquid nitrogen and then stored in the laboratory at -80˚C until further used.

### Procurement of early stage zebrafish embryos

Six zebrafish couples were paired to obtain embryos. A total of 390 embryos were obtained and pooled in glass Petri dishes (≈30 embryos per dish) with clean water. Embryos suffering malformations or with retarded development were removed. Three groups with around 100 embryos were collected after 2, 8 and 30 hours post fertilisation (hpf), immersed in TRizol®

(Invitrogen, Carlsbad, California, USA) and maintained at 4˚C to proceed immediately with RNA extraction.

## Histological analysis and staging

After 24 hours in the fixative samples of hormone treatment experiment were dehydrated in a graded series of ethanol in a Leica ASP 300 tissue processor (Leica Biosystems, Wetzlar, Germany) and embedded in paraffin. 5 μm thickness sections were cut in a 2065 Supercut microtome and stained with hematoxylin/eosin using the Leica Autostainer XL workstation and mounted with the aid of the Leica CV 5030 workstation. The slides were microscopically examined under an Olympus BX61 light microscope (Tokyo, Japan).

## RNA extraction and cDNA synthesis

Total RNA was extracted from 50–100 mg of tissue, or whole body in the case of exposed zebrafish larvae, using TRIzol® (Invitrogen) and following the manufacturer´s instructions. Obtained RNA quality was checked in an Agilent RNA 6000 Nano Kit Bioanalyzer (Agilent Technologies, Santa Clara, California, USA).

First-strand cDNA was synthesizsd using the SuperScript™ First-Strand Synthesis System for retrotranscription (Invitrogen) in the 2720 Applied Biosystems Thermal Cycler (Life Technologies). It was performed according to manufacturer's instructions using a maximum of 2 μg total RNA in a reaction volume of 20 μl (100 ng/μL final theoretical cDNA concentration). The concentration of single stranded cDNA (ssDNA) was quantified by fluorescence in the Synergy HT Multi-Made Microplate Reader (BioTek) using Quant-iT™ OliGreen® ssDNA Assay Kit (Invitrogen, Life Technologies). The quantification was run in triplicates, in a reaction volume of 100 μl, with a theoretical cDNA concentration of 0.2 ng/μL. The fluorescence was measured at 485/20 nm excitation and 528/20 nm emission wavelengths. Real cDNA concentration was calculated using the high-range standard curve according to the manufacturer's instructions. Once cDNA concentration was calculated, the exact amount of cDNA loaded in the qPCR reactions was calculated adjusting the dilution used for each gene.

## *gtf3aa* and *gtf3ab* transcription pattern in zebrafish and tilapia

*gtf3aa* and *gtf3ab* mRNA fragments were designed for tilapia an zebrafish using sequences obtained from Ensembl and NCBI (Table 1). They were designed in exon-exon boundaries to avoid amplification of genomic DNA and amplified using End-point RT-PCR employing

**Table 1. Primer sequences used for the End-Point PCR and qPCR analysis of different target genes in zebrafish and tilapia.** Amplified fragment size in bp and PCR annealing temperature (˚C) are indicated.

|  | Species | Gene | NCBI accession number | Forward sequence (5'-3') | Reverse sequence (5'-3') | bp | ˚C |
|---|---|---|---|---|---|---|---|
| *PCR* | *O. niloticus* | *gtf3aa* | XM_003454117 | ATCTGTTCGTTTAGCGGCTGCT | GTATTTCCTCGGCTCCAGGC | 276 | 60 |
|  |  | *gtf3ab* | XM_00344354 | CACCCGCTACCAACTCACCA | CTGATGGACTCGGGCAATGT | 133 | 60 |
|  | *D. rerio* | *gtf3aa* | NM_001003866 | CACACTCAGCTTCTACCTTTCT | GGTCTCACAAGAGTAGCCTTTAT | 114 | 59 |
|  |  | *gtf3ab* | NM_001089544 | TTGCATGTGGAGACTGTGAGAAGA | CTGACTGAACACAGGTAAGGCTT | 100 | 60 |
| *qPCR* | *D. rerio* | *amh* | NM_001007779 | AGGCTCAGTACCGTTCAGTGTT | TCTTCATCAGCTCTCGCTGCT | 100 | 59 |
|  |  | *actb* | NM_131031 | CATCTATGAGGGTTACGCTCTT | TCTCTTTCGGCTGTGGTGG | 129 | 58.8 |
|  |  | *cyp19a1a* | NM_131154 | CTCAATGAGCACGATCTGCTT | CTCCTGAGCATCTCTTTTGTG | 129 | 57.9 |
|  |  | *dmrt1* | NM_205628 | CAGGTTCCTCGTGCCAACA | GGGACGGTTTCCTGATGGA | 173 | 58.8 |
|  |  | *gtf3aa* | XM_003454117 | CATCCCGTCTGAGTGGCTACA | CTACACTAAGAAGGGCTCTAATAGG | 224 | 61 |
|  |  | *gtf3ab* | XM_00344354 | TAGGAAGCTGCATGAAGGTTA | ACATGGAAGGTTTACTCTGTG | 115 | 55 |

0.8 mM primers (Table 1). Properties of designed primers were checked employing the IDT OligoAnalyzer Tool (https://eu.idtdna.com/calc/analyzer) and purchased from Eurofins MWG.

Amplifications were run with commercial Taq DNA Polymerase, recombinant Kit and 100 mM dNTP Mix (Invitrogen) for 35 cycles in a 2720 Thermal Cycler (Applied Biosystems, Carlsbad, California, USA). PCR procedure was as follows: 94˚C for 2 minutes, and 35 cycles of denaturation at 94˚C for 30 seconds, annealing step (temperature for each primer set in Table 1) for 30 seconds and elongation at 72˚C for 30 seconds, and a final step at 72˚C for 8 minutes. PCR end-products were visualised in 1.5% agarose gels stained with ethidium bromide.

## Quantitative RT-PCR (qPCR) analysis

Sequences for *Danio rerio actb*, *amh*, *cyp19a1a*, *dmrt1*, *gtf3aa* and *gtf3ab* were obtained from NCBI (Table 1). Transcription levels were determined using SYBR Green PCR Master Mix (Roche). Optimal concentrations of primers (12.5 mM) and theoretical sample concentrations of 8 ng/μL were used for each gene. Samples were run in triplicates in a 7300 PCR thermal cycler (Applied Biosystems) using a final reaction volume of 20 μL, containing 2 μL of appropriately diluted sample. Reaction conditions were as follows: 2 min at 50˚C, 10 min at 95˚C, followed by 40 cycles of 15 s at 95˚C and annealing step of 60 s at appropriate temperature (Table 1). Amplification reaction was followed by a dissociation stage to obtain a dissociation curve, which would allow checking the specificity of each primer set and ensuring that only the specific transcript was amplified.

Transcription levels were normalised taking into account the amount of cDNA loaded for each sample as measured by fluorescence. All gene transcription results were normalised with the amount of cDNA charged in the qPCR according to Rojo-Bartolomé et al., 2016 using an adapted ΔCT formula for relative quantification of each gene (*RQ*) with efficiency correction (*E*):

$$E = \left[10^{-1/m}\right] - 1 \tag{1}$$

*m* being the slope of the standard curve of the qPCR reaction.

$$RQ = Log_2\left[\frac{(1 + Efficiency)^{-\Delta CT}}{ngcDNA}\right] \tag{2}$$

Where

$$\Delta CT = CT\ sample - CT\ interplate\ internal\ control \tag{3}$$

For each gene a pool with cDNA of all measured samples was produced and different dilutions were used to generate an amplification standard curve. Then, one of these dilutions in the middle of the curve was included as reference in all measured plates (3 in total) for each gene (interplate internal control).

In the hormone exposure experiment final RQ values were obtained as follows:

$$RQ\ sample - \overline{RQ}\ corresponding\ reference\ group \tag{4}$$

Reference group in each case was chosen according to the nature of the studied gene since exposure control groups (ET) contained both female and male individuals. This was done assuming lowest transcription levels for each gene in the corresponding reference group. In this way, MT group was selected as reference group in the case of female marker genes (*gtf3ab*

and *cyp19a1a*), ET group for *gtf3aa* and *actb*, and E group in the case of male marker genes (*amh* and *dmrt1*).

### Ethics statement

During the zebrafish egg production and exposure to hormones all animal manipulations conducted were authorised by competent regional authorities after the evaluation and approval of all protocols by the Ethics and Animal Welfare Commission of the University of the Basque Country (CEEA/337-2/2014). The methods in this research were carried out in accordance with the approved guidelines.

Furthermore, zebrafish and tilapia fish species were anaesthetised in a saturated ethyl 4-aminobenzoate (Fluka, Teinheim, Germany) water bath following the protocol authorised by the ethics commission of the University of the Basque Country (CEEA/152/2010). The methods in this section were carried out in accordance with the approved guidelines.

### Statistical analysis

The statistical analyses were undertaken using SPSS (SPSS Inc., Chicago, Illinois). Data failed in normality and variance equality after applying the Shapiro-Wilk (n<30) test and Levene's test, both at a 0.05 significance level (p<0.05). Significant differences between replicates were thus established using the non-parametric Mann-Whitney test and differences between experimental groups were then evaluated using the non-parametric Kruskal-Wallis test. In all the cases, significant differences were established at p< 0.05.

## Supporting information

**S1 Fig. *gtf3aa* and *gtf3ab* transcript levels in different tilapia tissues.** Agarose gel electrophoresis after PCR with cDNA generated from tilapia ovary (O), testis (T), gill (G) and eye (E). Fragments were around 280 nucleotides in length for *gtf3aa* and around 130 for *gtf3ab*. An unspecific amplicon can be observed in the *gtf3ab* gels that does not mask the specific ovarian amplicon (arrow). Ø = no template control; L = Standard 50 bp (Invitrogen).
(DOC)

**S2 Fig. Transcript levels of genes related to ovarian (*gtf3ab* and *cyp19a1a*) and testicular (*amh* and *dmrt1*) differentiation in hormone treated zebrafish after 61 days of exposure.** The experimental groups after 61 days of exposure were: 17β-estradiol (E61), 17α-methyltestosterone (MT61) and ethanol control group which was separated in female and male (ET61_F vs ET61_M) considering *gtf3ab* transcription levels. Box plots represent the data within the 25th and 75th percentiles, with the median indicated by a line, and top and bottom whiskers indicating the minimum and maximum values (12 individuals per treatment group with 5 individuals in ET61_F and 7 in ET61_M). Different letters indicate significant differences between groups (Kruskal-Wallis, p<0.05).
(DOC)

**S3 Fig. Fig generated in Ensembl with the data obtained from different RNA-seq experiments in zebrafish and that depicts transcript levels in different tissues superimposed on the region of the genome where *gtf3aa* is located.** Notice that *mtif3* and *gtf3aa* are always similarly transcribed in all depicted tissues and developmental stages.
(DOC)

**S4 Fig. *gtf3ab* gene transcription profiles in zebrafish tissues and along development.** Fig generated in Ensembl with the data obtained from different RNA-seq experiments in zebrafish

and that depicts transcript levels in different tissues superimposed on the region of the genome where *gtf3ab* is located. Notice *gtf3ab* is only expressed in ovary, whole female larvae and in the very early developmental stages.
(DOC)

**S5 Fig. Synteny analysis of gtf3ab gene using zebrafish genome as reference (Genomicus v96.01 webtool).**
(DOCX)

**S1 Table. Ensembl reference IDs and locations for each of the genes neighbouring *gtf3a* orthologs in animal genomes studied in the synteny analysis.**
(DOC)

**S2 Table. Zebrafish body length (mm) at days 26 and 61 in the experiment.**
(DOC)

**S3 Table. Protein sequences deduced from the cds sequences belonging to different *gtf3a* ortholog genes, with length and deduced molecular weight.** The sequences presented here were the ones used to produce the phylogenetic tree in S1 Fig. Sequence underlined in yellow show the conserved initial sequence in all the oocyte specific Gtf3ab proteins of fish and the Xenopus oocytic protein. The sequence in blue shows the last of the C2H2 Zn finger domains of all the Gtf3as. In green the conserved transcription activation KRSLAS domain (KRSLAShLsGYPPK), necessary for transcriptional activation of 5S rRNA is shown. In teleostean proteins this is only found in Gtf3aa-s.
(DOC)

**S4 Table. RNA-seq results of pirarucu transcriptome.** Comparison of the read-counts for gtf3a transcripts across tissues and sexes in pirarucu as conducted by Vialle and de Souza et al., 2018. TMP. Transcrips per Million.
(DOC)

## Acknowledgments

We are thankful to BREEN, Ltd., (NER group, Hondarribia, Spain) for providing tilapia samples.

## Author Contributions

**Conceptualization:** Iratxe Rojo-Bartolomé, Jorge Estefano Santana de Souza, Oihane Diaz de Cerio, Ibon Cancio.

**Data curation:** Iratxe Rojo-Bartolomé, Jorge Estefano Santana de Souza, Oihane Diaz de Cerio.

**Funding acquisition:** Ibon Cancio.

**Investigation:** Iratxe Rojo-Bartolomé.

**Methodology:** Iratxe Rojo-Bartolomé.

**Supervision:** Ibon Cancio.

**Writing – original draft:** Iratxe Rojo-Bartolomé.

**Writing – review & editing:** Oihane Diaz de Cerio, Ibon Cancio.

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
