## [Decision Letter · Decision Letter 0]

8 Nov 2019

PONE-D-19-26233

Duplication and subfunctionalization of the general transcription factor IIIA (gtf3a) gene in teleost genomes, with ovarian specific transcription of gtf3ab

PLOS ONE

Dear Dr Cancio,

Thank you for submitting your manuscript to PLOS ONE. After careful consideration, we feel that it has merit but does not fully meet PLOS ONE’s publication criteria as it currently stands. Therefore, we invite you to submit a revised version of the manuscript that addresses the points raised during the review process.

We would appreciate receiving your revised manuscript by Dec 23 2019 11:59PM. To enhance the reproducibility of your results, we recommend that if applicable you deposit your laboratory protocols in protocols.io, where a protocol can be assigned its own identifier (DOI) such that it can be cited independently in the future. For instructions see: http://journals.plos.org/plosone/s/submission-guidelines#loc-laboratory-protocols

We look forward to receiving your revised manuscript.

Kind regards,

Gao-Feng Qiu

Academic Editor

PLOS ONE

1. PLOS ONE now requires that authors provide the original uncropped and unadjusted images underlying all blot or gel results reported in a submission’s figures or Supporting Information files. This policy and the journal’s other requirements for blot/gel reporting and figure preparation are described in detail at https://journals.plos.org/plosone/s/figures#loc-blot-and-gel-reporting-requirements and https://journals.plos.org/plosone/s/figures#loc-preparing-figures-from-image-files. When you submit your revised manuscript, please ensure that your figures adhere fully to these guidelines and provide the original underlying images for all blot or gel data reported in your submission. See the following link for instructions on providing the original image data: https://journals.plos.org/plosone/s/figures#loc-original-images-for-blots-and-gels.

2. Thank you for including the following funding information within your acknowledgements section of your manuscript; "This work has been funded through research projects of MINECO (AGL2012-33477 and AGL2015-63936_R), Basque-Government (PhD fellowship to IRB, S-PE13UN101 & IT810-13). The funders had no role in study design, data collection and analysis, decision to publish, or preparation of the manuscript. "

Reviewers' comments:

Reviewer's Responses to Questions

**Comments to the Author**

1. Is the manuscript technically sound, and do the data support the conclusions?

Reviewer #1: Partly

Reviewer #2: Yes

2. Has the statistical analysis been performed appropriately and rigorously? 

Reviewer #1: N/A

Reviewer #2: Yes

3. Have the authors made all data underlying the findings in their manuscript fully available?

Reviewer #1: Yes

Reviewer #2: Yes

4. Is the manuscript presented in an intelligible fashion and written in standard English?

Reviewer #1: Yes

Reviewer #2: Yes

5. Review Comments to the Author

Reviewer #1: “Duplication and subfunctionalization of the general transcription factor IIIA (gtf3a) gene in teleost genomes, with ovarian specific transcription of gtf3ab” by Ibon Cancio, et al.

They manuscript was attempted to characterize zebrafish two gtf3 duplicates ( Gtf3a and gtf3b), and their functions in oocyte development. The topic is interesting. However, the present data is not sufficient to support the conclusion.

1. Histological images (Fig.4) are not clear. Higher magnifications and high quality figures are needed to clearly show the morphological changes of gonadal tissues and cells.

2. It is hard to understand the methods and results of transcription quantification analyses.

a) Is the conventional PCR conventional RT-PCR (Reverse transcription PCR)?

b) Is the qPCR analysis different from the most publications? For example, ΔCT= mean Ct (sample)-mean Ct (internal reference); ΔΔCT=ΔCt (treated)- ΔCt (control). In the manuscript, it was unclear what was plate internal control, what was reference group, and how to calculate the differences between groups?

c) The figures and legends are obscured (Fig.3 and Fig.5). For example, two figures did not show how to compare with the control or /and internal control (beta actin); Y axis was marked as RQ. In Fig.3 legend: “Amplified fragments were around 100 nucleotides in size”, the authors didn’t predicate (calculate) the exact size?.

Reviewer #2: In this study, the authors identified two gtf3a paralog in teleost species and characterized their divergence function on fish oogenesis. By phylogenetic and syntenic analysis, they found the duplication of gtf3a is result from the teleost specific genome duplication and the gtf3ab evolved into the conserved genomic distribution pattern with gtf3a of other vertebrates. Further, the gtf3ab was verified that it took part in the fish oogenesis based on the expression of different tissues, early stage and exposure treatment. All these results suggest that the gtf3ab is a useful marker of gonad feminisation and intersex condition in fish species. Anyway, the interpretation of the results is sound for the most part, and gives enough proof to verify their results. My major concern is the discussion part looks a bit messy and include some data that are not mentioned in the results. Some descriptions can move to the result and correspondingly some conclusions would be strengthened with some reorganization and more thorough editing by a native English reader with some expertise in the science presented. I have no specific comments.

6. PLOS authors have the option to publish the peer review history of their article (what does this mean?). If published, this will include your full peer review and any attached files.

Reviewer #1: No

Reviewer #2: No

---

## [Author Response · Author response to Decision Letter 0]

10 Dec 2019

Dear editors of PLOS one!!

Thanks to your comments or our manuscript and to those of both reviewers, which we think are very positive about the manuscript. We have carefully addressed the recommendations of the editors and suggestions of the reviewers as you will be able to observe in the ms marked in track changes.

Regarding the comments of the editors: 1.- we have carefully looked at the file naming, 2.- we provide original uncropped images, 3.- the mention to data not shown has been removed as it was not necessary at all, and 4.- ORCID ID of corresponding author has been incorporated.

We have removed any mention to funding bodies form the Acknowledgements sections. In this sense, we would like to include the following text as Funding Statement: 

“This work has been funded through research projects of MINECO (AGL2012-33477 and AGL2015-63936_R), Basque-Government (PhD fellowship to IRB, S-PE13UN101 & IT810-13). The funders had no role in study design, data collection and analysis, decision to publish, or preparation of the manuscript.” 

As to the Reviewers' comments:

Reviewer #1: “Duplication and subfunctionalization of the general transcription factor IIIA (gtf3a) gene in teleost genomes, with ovarian specific transcription of gtf3ab” by Ibon Cancio, et al.

They manuscript was attempted to characterize zebrafish two gtf3 duplicates (gtf3a and gtf3b), and their functions in oocyte development. The topic is interesting. However, the present data is not sufficient to support the conclusion.

1.- Histological images (Fig.4) are not clear. Higher magnifications and high quality figures are needed to clearly show the morphological changes of gonadal tissues and cells.

Sorry, but in our view micrographs and the way they discussed are very clear. Testis can be seen in some individuals and ovary in some others. The only function of such micrographs is to illustrate that all fish treated with MT produced testis at day 61 and the ones treated with E2 ovary. In ET controls there were males with testes and females with ovary. 

We have not observed any histopathological alteration (and we do not mention anything in that sense) so we have not felt the need to show higher mag micrographs.

2. It is hard to understand the methods and results of transcription quantification analyses.

a) Is the conventional PCR conventional RT-PCR (Reverse transcription PCR)?

Yes “conventional PCR” meant End-Point RT-PCR (as explained in the materials and methods). We used the word conventional to differentiate it from qPCR, in this case quantitative RT-PCR. “Conventional PCR” is widely used for this purpose is widely in scientific publications. (Science direct provides 8,252 hits). However, we have modified the text as suggested by the reviewer and used End-Point RT-PCR and qPCR for both cases in the amended text.

b) Is the qPCR analysis different from the most publications? For example, ΔCT= mean Ct (sample)-mean Ct (internal reference); ΔΔCT=ΔCt (treated)- ΔCt (control). In the manuscript, it was unclear what was plate internal control, what was reference group, and how to calculate the differences between groups?

b) The manuscript mentions in lines 394-400 (of the ms in track changes) and in materials and methods that we quantified the cDNA amount in our samples. We do not use any housekeeping gene as we prefer another approach through the exact quantification of cDNA in each sample using quantIT. Working with fish along development, or with gonads that change molecularly and functionally so much during time and through gametogenesis, we consider that there are no genes “keeping any house”. In fact, we provide information on actb (widely used housekeeping gene) as another target gene in figure 5. When using actb as housekeeping gene in the traditional way in this experiment, there were changes in two genes in relation only to the MT60 group in figure 5. There has been a “fashion” until very recently to publish housekeeping validation papers, that is mostly over now. We always load the same quantity of cDNA per sample in our PCRs, and for that reason, we quantify exactly the amount of cDNA through fluorescence. This was (and is) long explained in the manuscript.

Regarding the ΔΔCt we explain our approach in length in lines 538 to 558 (in ms with track changes). In results from figure 5 our control (reference) group is the vehicle (ethanol) treated group. We are comparing sex related gene expression profiles with 2 treated groups. One of them, as a consequence of treatment develops testis, the other one ovaries. Male related genes show high expression in testes, low in ovaries, and for other genes it is the other way around. Well, in the Reference group, we have individuals that are female and individuals that are male (and as we are working with whole larvae, we do not know which ones are females and which males). Obviously, we can never use this for the ΔΔCT, so instead we use for each gene the group in which the transcription levels are at their lowest. 

c) The figures and legends are obscured (Fig.3 and Fig.5). For example, two figures did not show how to compare with the control or /and internal control (beta actin); 

Y axis was marked as RQ. 

c) It could be a misunderstanding but the gels of Fig 3 do not have a quantitative purpose. They just show the absence of gene transcription in some tissues. The legend of the figure has been adapted as follows to clarify this point: Fig 3. “A) Agarose gel electrophoresis showing presence vs absence of gtf3aa and gtf3ab amplicons after End-Point RT-PCR in organs of three male and three female individuals.” Regarding beta actin we have explained everything above.

Regarding calculation of RQ everything is explained in materials and methods section (lines 538-558 of track changes marked version). An explanatory sentence has been added in line 547 to understand what was meant by internal plate control (in fact, interplate control… changed in formula 3, line 546). There was an error in formula number 4 that has been corrected (- instead of +), line 552. 

In figure 3C there is no “treatment control group”. 

In Fig.3 legend: “Amplified fragments were around 100 nucleotides in size”, the authors didn’t predicate (calculate) the exact size?.

Exact size of amplified fragments was already provided in Table 1. It is 100 bp for gtf3aa and 114 for gtf3ab, that is why we wrote that the amplified fragments were around around 100 nucleotide size in figure legend. It is now specified more clearly in the legend. 

Reviewer #2: In this study, the authors identified two gtf3a paralog in teleost species and characterized their divergence function on fish oogenesis. By phylogenetic and syntenic analysis, they found the duplication of gtf3a is result from the teleost specific genome duplication and the gtf3ab evolved into the conserved genomic distribution pattern with gtf3a of other vertebrates. Further, the gtf3ab was verified that it took part in the fish oogenesis based on the expression of different tissues, early stage and exposure treatment. All these results suggest that the gtf3ab is a useful marker of gonad feminisation and intersex condition in fish species. Anyway, the interpretation of the results is sound for the most part, and gives enough proof to verify their results. My major concern is the discussion part looks a bit messy and include some data that are not mentioned in the results. Some descriptions can move to the result and correspondingly some conclusions would be strengthened with some reorganization and more thorough editing by a native English reader with some expertise in the science presented. I have no specific comments.

Thanks to reviewer and we are happy that she/he was satisfied by the science behind our manuscript. 

We have tried to clarify the discussion section bringing a few statements to the results part. We believe that the reviewer was refereeing to our calls in the Discussion section to the supplementary figures and tables. These parts have been transported to the results section.

The English has been checked by a native speakers in our marine station who pointed out to some minor changes introduced in the new ms.

---

## [Decision Letter · Decision Letter 1]

27 Dec 2019

Duplication and subfunctionalisation of the general transcription factor IIIA (gtf3a) gene in teleost genomes, with ovarian specific transcription of gtf3ab

PONE-D-19-26233R1

Dear Dr. Cancio,

We are pleased to inform you that your manuscript has been judged scientifically suitable for publication and will be formally accepted for publication once it complies with all outstanding technical requirements.

With kind regards,

Gao-Feng Qiu

Academic Editor

PLOS ONE

Additional Editor Comments (optional):

Reviewers' comments:

Reviewer's Responses to Questions

**Comments to the Author**

1. If the authors have adequately addressed your comments raised in a previous round of review and you feel that this manuscript is now acceptable for publication, you may indicate that here to bypass the “Comments to the Author” section, enter your conflict of interest statement in the “Confidential to Editor” section, and submit your "Accept" recommendation.

Reviewer #1: All comments have been addressed

Reviewer #2: All comments have been addressed

2. Is the manuscript technically sound, and do the data support the conclusions?

Reviewer #1: Yes

Reviewer #2: Yes

3. Has the statistical analysis been performed appropriately and rigorously? 

Reviewer #1: Yes

Reviewer #2: Yes

4. Have the authors made all data underlying the findings in their manuscript fully available?

Reviewer #1: Yes

Reviewer #2: Yes

5. Is the manuscript presented in an intelligible fashion and written in standard English?

Reviewer #1: Yes

Reviewer #2: Yes

6. Review Comments to the Author

Reviewer #1: The data would be much taugh/strong enough if the transition images are provided during treatment.

Reviewer #2: The authors have addressed my concerns in the first round review. The paper looks ready now and I have no comments again.

7. PLOS authors have the option to publish the peer review history of their article (what does this mean?). If published, this will include your full peer review and any attached files.

Reviewer #1: No

Reviewer #2: No

---

## [Editor Report · Acceptance letter]

3 Jan 2020

PONE-D-19-26233R1 

Duplication and subfunctionalisation of the general transcription factor IIIA (*gtf3a*) gene in teleost genomes, with ovarian specific transcription of *gtf3ab*

Dear Dr. Cancio:

I am pleased to inform you that your manuscript has been deemed suitable for publication in PLOS ONE. Congratulations! Your manuscript is now with our production department. 

With kind regards,

on behalf of

Prof. Gao-Feng Qiu 

Academic Editor

PLOS ONE